# Incorporating Prior Knowledge into Neural Networks through an Implicit Composite Kernel

**Ziyang Jiang**                                           *ziyang.jiang@duke.edu*
*Department of Civil and Environmental Engineering*
*Duke University*

**Tongshu Zheng**                                      *tongshuzheng92@gmail.com*
*Division of Natural and Applied Science*
*Duke Kunshan University*

**Yiling Liu**                                                *yiling.liu@duke.edu*
*Program in Computational Biology and Bioinformatics*
*Duke University School of Medicine*

**David Carlson**                                         *david.carlson@duke.edu*
*Department of Civil and Environmental Engineering*
*Department of Biostatistics and Bioinformatics*
*Department of Computer Science*
*Duke University*

**Reviewed on OpenReview:** *https://openreview.net/forum?id=HhjSaluWVe*

## Abstract

It is challenging to guide neural network (NN) learning with prior knowledge. In contrast, many known properties, such as spatial smoothness or seasonality, are straightforward to model by choosing an appropriate kernel in a Gaussian process (GP). Many deep learning applications could be enhanced by modeling such known properties. For example, convolutional neural networks (CNNs) are frequently used in remote sensing, which is subject to strong seasonal effects. We propose to blend the strengths of NNs and the clear modeling capabilities of GPs by using a composite kernel that combines a kernel implicitly defined by a neural network with a second kernel function chosen to model known properties (e.g., seasonality). We implement this idea by combining a deep network and an efficient mapping function based on either Nyström approximation or random Fourier features, which we call Implicit Composite Kernel (ICK). We then adopt a sample-then-optimize approach to approximate the full GP posterior distribution. We demonstrate that ICK has superior performance and flexibility on both synthetic and real-world datasets including a remote sensing dataset. The ICK framework can be used to include prior information into neural networks in many applications.

## 1 Introduction

In complex regression tasks, input data often contains *multiple sources of information*. These sources can be presented in both high-dimensional (e.g. images, audios, texts, etc.) and low-dimensional (e.g. timestamps, spatial locations, etc.) forms. A common approach to learn from high-dimensional information is to use neural networks (NNs) (Goodfellow et al., 2016; LeCun et al., 2015), as NNs are powerful enough to capture the relationship between complex high-dimensional data and target variables of interest. In many areas, NNs are standard practice, such as the dominance of Convolutional Neural Networks (CNNs) for image analysis (Jiang et al., 2022; Zheng et al., 2021; 2020). In contrast, for low-dimensional information, we usually have

some prior knowledge on how the information relates to the predictions. As a concrete example, consider a remote sensing problem where we predict ground measurements from satellite imagery with associated timestamps. *A priori*, we expect the ground measurements to vary periodically with respect to time between summer and winter due to seasonal effects. We would typically use a CNN to capture the complex relationship between the imagery and the ground measurements. In this case, we want to guide the learning of the CNN with our prior knowledge about the seasonality. This is challenging because knowledge represented in NNs pertains mainly to correlation between network units instead of quantifiable statements (Marcus, 2018).

Conversely, Gaussian processes (GPs) have been used historically to incorporate relevant prior beliefs by specifying the appropriate form of its kernel (or covariance) function (Bishop & Nasrabadi, 2006; Williams & Rasmussen, 2006). One approach to modeling multiple sources of information is to assign a relevant kernel function to each source of information respectively and combine them through addition or multiplication, resulting in a *composite kernel function* (Duvenaud, 2014). This formulation means that specifying a kernel to match prior beliefs on one source of information is straightforward. Such composite kernel learning techniques are extensively used in many application areas such as multi-media data (McFee et al., 2011), neuroimaging (Zhang et al., 2011), spatial data analysis, and environmental data analysis (Kim et al., 2005; Petelin et al., 2013). In view of the clear modeling capabilities of GP, it is desirable to examine how a NN could be imbued with the same modeling ease.

In recent years, researchers have come up with a variety of methods to incorporate prior knowledge into NNs. These efforts can be broken into many categories, such as those that add prior information through loss terms like physics-informed NNs (Lagaris et al., 1998; Moseley et al., 2020). Here, we focus on the major category of those methods that build integrated models of NNs and GPs with various structures (Van der Wilk et al., 2017; Wilson et al., 2016b; 2011). Related to our proposed methodology, Pearce et al. (2020) exploited the fact that a Bayesian neural network (BNN) approximates a GP to construct additive and multiplicative kernels, but they were limited to specific predefined kernels. Matsubara et al. (2020) then resolved this limitation by constructing priors of BNN parameters based on the ridgelet transform and its dual, but they did not explicitly show how their approach works for data with multiple sources of information. To our knowledge, no existing approach allows a modeler to choose any appropriate kernel over multiple sources.

We address this limitation by presenting a simple yet novel Implicit Composite Kernel (ICK) framework, which processes high-dimensional information using a kernel implicitly defined by a neural network and low-dimensional information using a chosen kernel function. The low-dimensional kernels are mapped into the neural network framework to create a straightforward and simple-to-learn implementation. Our key results and contributions are:

- We analytically show our ICK framework, under reasonable assumptions, is approximately equivalent to sampling from a Gaussian process regression (GPR) model with a composite kernel *a priori*.

- We adopt a sample-then-optimize procedure to ICK to approximate the full posterior distribution of a GP with a composite kernel.

- We show that ICK yields better performance on prediction and forecasting tasks, even with limited data.

- We show that ICK can flexibly capture the patterns of the low-dimensional information without bespoke pre-processing procedures or complex NN structures.

Based on these contributions, we believe ICK is useful in learning from complex *hybrid* data with prior knowledge, especially in remote sensing and spatial statistics.

## 2   Related Work

**Equivalence between NNs and GPs**   The equivalence between GPs and randomly initialized single-layer NNs with infinite width was first shown by Neal (1996). With the development of modern deep learning, researchers further extended this relationship to deep networks (Lee et al., 2017; Matthews et al., 2018) and convolutional neural networks (CNNs) (Garriga-Alonso et al., 2018; Novak et al., 2018). This relationship is crucial for showing the resemblance between GPR and our ICK framework, which is discussed in Section 4.1.

**NNs with prior knowledge**   As mentioned before, one approach to equip NNs with prior knowledge is to modify the loss function. For example, Lagaris et al. (1998) solved differential equations (DEs) using NNs by setting the loss to be a function whose derivative satisfies the DE conditions. Another approach is to build integrated models of NNs and kernel-based models. For example, Wilson et al. (2011) implemented a regression network with GP priors over latent variables and made inference by approximating the posterior using Variational Bayes or sampling from the posterior using Gibbs sampling scheme. Garnelo et al. (2018) introduced a class of neural latent variable models called Neural Processes (NPs) which are capable of learning efficiently from the data and adapting rapidly to new observations. In addition to these, various studies have explored the integration of Neural Networks (NNs) and kernel methods (Hinton & Salakhutdinov, 2007; Wilson et al., 2016a; Adlam et al., 2020). Our ICK framework fuses prior knowledge into NNs by modulating the learnt features using another set of features outputted from a kernel-based mapping, which can also be viewed as an integrated model of NNs and kernel machines.

**GP with composite kernels**   Composite kernel GPs are widely used in both machine learning (Duvenaud, 2014; Williams & Rasmussen, 2006) and geostatistical modeling (Datta et al., 2016; Gelfand & Schliep, 2016). GPR in geostatistical modeling is also known as *kriging* (Journel & Huijbregts, 1976; Krige, 1951), which serves as a surrogate model to replace expensive function evaluations. The inputs for a composite GP are usually low-dimensional (e.g. spatial distance) as GPs do not scale well with the number of samples for high-dimensional inputs (Bouhlel & Martins, 2019; Bouhlel et al., 2016). To overcome this issue, Pearce et al. (2020) and Matsubara et al. (2020) developed BNN analogues for composite GPs. Similar to these studies, our ICK framework can also be viewed as a simulation for composite GPs.

**GP for large datasets**   Since training and inference of exact GP scales $\mathcal{O}(N^3)$, either parallel computing Wang et al. (2019); Adlam et al. (2023) or kernel approximation are needed to scale GP to large datasets. Nyström low-rank matrix approximation (Drineas et al., 2005; Williams & Seeger, 2000) and Random Fourier Features (Rahimi & Recht, 2007; 2008) are two commonly used approximation methods. Building upon these concepts, several popular frameworks, such as sparse GPs (Snelson & Ghahramani, 2005; Titsias, 2009; Hensman et al., 2013), have been developed to facilitate GP inference on large datasets. In our research, we draw inspiration from these approximation methods and utilize them as *transformation functions* to project the kernel matrix into latent space representations, as discussed in Section 4.2.

## 3   Background

### 3.1   Problem Setup

To formalize the problem, we have a training data set which contains $N$ data points $\boldsymbol{X} = [\boldsymbol{x}_i]_{i=1}^N = [\boldsymbol{x}_1, \boldsymbol{x}_2, ..., \boldsymbol{x}_N]^T$ and the corresponding labels of these data points are $\boldsymbol{y} = [y_i]_{i=1}^N = [y_1, y_2, ..., y_N]^T$ where $y_i \in \mathbb{R}$. Each data point $\boldsymbol{x}_i = \{\boldsymbol{x}_i^{(1)}, \boldsymbol{x}_i^{(2)}, ..., \boldsymbol{x}_i^{(M)}\}$ is composed of information from $M$ different sources where the $m^{th}$ source of information of the $i^{th}$ data point is denoted as $\boldsymbol{x}_i^{(m)} \in \mathbb{R}^{D_m}$. Our goal is to learn a function $\hat{y}_i = f(\boldsymbol{x}_i) : \mathbb{R}^{D_1} \times \mathbb{R}^{D_2} \times ... \times \mathbb{R}^{D_M} \to \mathbb{R}$ which takes in a data point $\boldsymbol{x}_i$ and outputs a predicted value $\hat{y}_i$.

### 3.2   Composite GPs

A Gaussian process (GP) describes a distribution over functions (Williams & Rasmussen, 2006). A key property of GP is that it is completely defined by a mean function $\mu(\boldsymbol{x})$ and a kernel function $K(\boldsymbol{x}, \boldsymbol{x}')$ where $\boldsymbol{x}$ and $\boldsymbol{x}'$ represent different samples from the training dataset. The mean function $\mu(\boldsymbol{x})$ is often assumed to be zero for simplicity. In that case, the outcome function is

$$f(\boldsymbol{x}) \sim \mathcal{GP}\left(0, K(\boldsymbol{x}, \boldsymbol{x}')\right). \tag{1}$$

Any finite subset of random variables has a multivariate Gaussian distribution with mean $\boldsymbol{0}$ and kernel matrix $\boldsymbol{K}$ whose entries can be calculated as $\boldsymbol{K}_{ij} = K(\boldsymbol{x}_i, \boldsymbol{x}_j)$ where $1 \leq i, j \leq N$. In many situations, the full kernel function is built by a composite kernel by combining simple kernels through addition $K^{\text{comp}}(\boldsymbol{x}, \boldsymbol{x}') =$

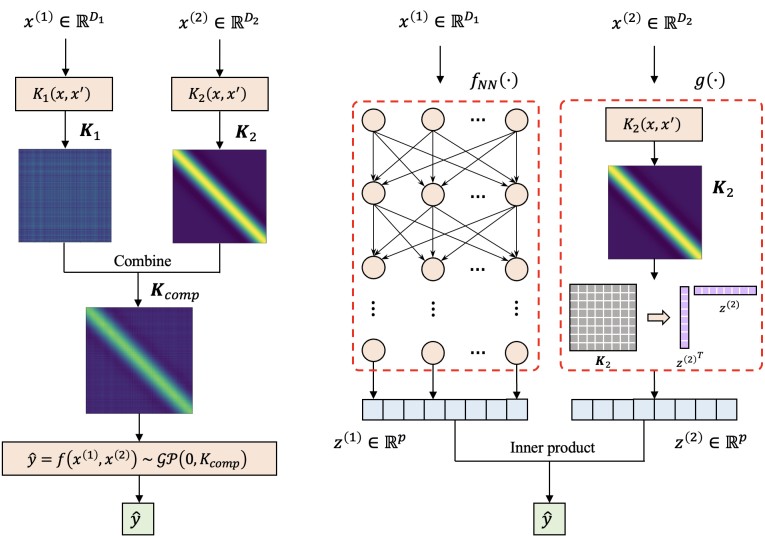

Figure 1: Given data containing 2 sources of information $\boldsymbol{x}^{(1)}$ and $\boldsymbol{x}^{(2)}$, we can process the data using either (**Left**) a composite Gaussian process regression (GPR) model or (**Right**) our ICK framework where $\boldsymbol{x}^{(1)}$ is processed with a neural network $f_{\text{NN}}(\cdot)$ and $\boldsymbol{x}^{(2)}$ is processed with $g(\cdot)$ where $g(\cdot)$ consists of a kernel function $K_2$ and some transformation which maps the kernel matrix $\boldsymbol{K}_2$ into the latent space.

$K_1(\boldsymbol{x}, \boldsymbol{x}') + K_2(\boldsymbol{x}, \boldsymbol{x}')$ or multiplication $K^{\text{comp}}(\boldsymbol{x}, \boldsymbol{x}') = K_1(\boldsymbol{x}, \boldsymbol{x}')K_2(\boldsymbol{x}, \boldsymbol{x}')$ (Duvenaud, 2014). A useful property that ICK exploits is that $K_1$ and $K_2$ can take different subparts of $\boldsymbol{x}$ as their inputs. For example, $K^{\text{comp}}(\boldsymbol{x}, \boldsymbol{x}') = K_1(\boldsymbol{x}^{(1)}, \boldsymbol{x}^{(1)'}) + K_2(\boldsymbol{x}^{(2)}, \boldsymbol{x}^{(2)'})$ or $K^{\text{comp}}(\boldsymbol{x}, \boldsymbol{x}') = K_1(\boldsymbol{x}^{(1)}, \boldsymbol{x}^{(1)'})K_2(\boldsymbol{x}^{(2)}, \boldsymbol{x}^{(2)'})$.

### 3.3 Correspondence between GPs and NNs

Neal (1996) proved that a single-hidden layer network with infinite width is *exactly equivalent* to a GP over data indices $i = 1, 2, ..., N$ under the assumption that the weight and bias parameters of the hidden layer are i.i.d. Gaussian with zero mean. This statement was then extended to deep NNs (Lee et al., 2017; Matthews et al., 2018) and convolutional NNs (Garriga-Alonso et al., 2018; Novak et al., 2018). Specifically, let $\boldsymbol{z} = f_{\text{NN}}(\boldsymbol{x}^{(1)}) : \mathbb{R}^{D_1} \to \mathbb{R}^p$ be the latent representation extracted from $\boldsymbol{x}^{(1)}$ where $p$ is the dimension of the extracted representation and $f_{\text{NN}}$ is a neural network with zero-mean i.i.d. parameters and continuous activation function $\phi$ which satisfies the linear envelope property

$$|\phi(u)| \le c + m|u| \quad \forall u \in \mathbb{R}, \tag{2}$$

if there exists $c, m \ge 0$. This includes many standard nonlinearities. The $k^{th}$ entry of this representation will converge in distribution to a NN-implied GP in the infinite width limit

$$f_{\text{NN}}(\boldsymbol{x}^{(1)})_k \xrightarrow{d} \mathcal{GP}(0, K^{\text{NN}}(\boldsymbol{x}^{(1)}, \boldsymbol{x}^{(1)'})). \tag{3}$$

Here $K^{\text{NN}}$ in Equation 3 denotes the covariance function of the equivalent GP as the width of $f_{\text{NN}}$ goes to infinity and can be computed numerically in a recursive manner (Lee et al., 2017). That is to say, the $k^{th}$ component $z_k$ of the representation extracted by the network has zero mean $\mathbb{E}_{p(\boldsymbol{\theta}^{(1)})}\left[z_{ik}^{(1)}\right] = 0$ for all $i = 1, 2, ..., N$ where $\theta$ represents the network parameters. The covariance between $z_{ik}^{(1)}$ and $z_{jk}^{(1)}$ for *different* data indices $i, j = 1, 2, ..., N$ can be approximated as $\text{cov}(z_{ik}^{(1)}, z_{jk}^{(1)}) = \mathbb{E}_{p(\boldsymbol{\theta}^{(1)})}\left[z_{ik}^{(1)}z_{jk}^{(1)}\right] \approx K^{\text{NN}}\left(\boldsymbol{x}_i^{(1)}, \boldsymbol{x}_j^{(1)}\right)$ where $\boldsymbol{x}_i^{(1)}$ and $\boldsymbol{x}_j^{(1)}$ are the corresponding inputs in case the network width is finite.

## 4 Implicit Composite Kernel

We show the structure of a composite GPR model and our ICK framework in Figure 1. To make the illustration clear, we limit ourselves to data with information from 2 different sources $\boldsymbol{x} = \{\boldsymbol{x}^{(1)}, \boldsymbol{x}^{(2)}\}$ where $\boldsymbol{x}^{(1)}$ is high-dimensional and $\boldsymbol{x}^{(2)}$ is low-dimensional (i.e. $D_1 \gg D_2$) with some known relationship with the target $y$. We are inspired by composite GPR, which computes 2 different kernel matrices $\boldsymbol{K}_1$ and $\boldsymbol{K}_2$ and

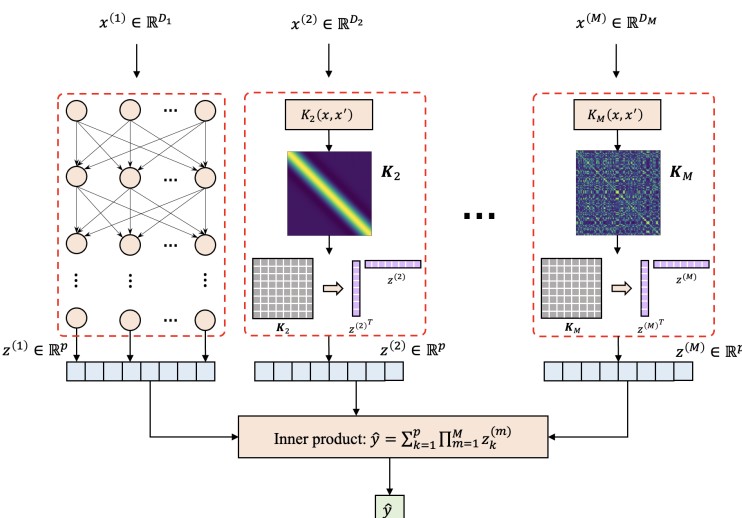

Figure 2: Given data containing $M$ sources of information $\boldsymbol{x} = \left\{x^{(1)}, x^{(2)}, ..., x^{(M)}\right\}$, we can process the data using our ICK framework where high-dimensional information (e.g. $x^{(1)}$ in the figure) is processed using a neural network and low-dimensional information (e.g. $x^{(2)}$ in the figure) is processed using a kernel function and some transformation which maps the kernel matrix into the latent space.

then combines them into a single composite kernel matrix $\boldsymbol{K}^{\text{comp}}$. However, as discussed before, it is more suitable to use a NN to learn from the high dimensional information $\boldsymbol{x}^{(1)}$. In our ICK framework, we process $\boldsymbol{x}^{(1)}$ with a NN $f_{\text{NN}}(\cdot) : \mathbb{R}^{D_1} \to \mathbb{R}^p$ and $\boldsymbol{x}^{(2)}$ with a mapping $g(\cdot) : \mathbb{R}^{D_2} \to \mathbb{R}^p$, which consists of a kernel function $K_2$ followed by a kernel-to-latent-space transformation (described in Section 4.2), resulting in two latent representations $\boldsymbol{z}^{(1)}, \boldsymbol{z}^{(2)} \in \mathbb{R}^p$. Then, we make a prediction $\hat{y}$ by doing an *inner product* between these two representations $\hat{y} = f_{\text{NN}}\left(\boldsymbol{x}^{(1)}\right) \cdot g\left(\boldsymbol{x}^{(2)}\right)$. Finally, the parameters of both the NN and the kernel function are learned via gradient-based optimization methods (Bottou et al., 2018).

Besides the formulation in Figure 1, ICK can also process data $\boldsymbol{x} = \left\{x^{(1)}, x^{(2)}, ..., x^{(M)}\right\}$ with $M > 2$ sources of information as shown in Figure 2. Here $K_1, ..., K_M$ represent different types of kernels with trainable parameters. The final prediction is calculated by a chained inner product of all extracted representations $\hat{y} = \sum_{k=1}^{p} \prod_{m=1}^{M} z_k^{(m)}$.

In the sections below, we first show the relationship between ICK and a composite GPR model with a multiplicative kernel *a priori* in Section 4.1, which is used to **motivate the model form**. We then show how we implement the kernel-to-latent-space transformation in Section 4.2. To enable a full GP posterior *approximation* for ICK, we will provide insight into how we derive uncertainty estimates from ICK. This can be achieved either through direct variance calculation (Section 4.3.1) or through the utilization of an ensemble approach (Section 4.3.2).

## 4.1 Resemblance between Composite GPR and ICK

We will analytically prove the following theorem for data with information from 2 different sources $\boldsymbol{x} = \left\{\boldsymbol{x}^{(1)}, \boldsymbol{x}^{(2)}\right\}$ for clarity, and we note this theorem can be straightforwardly extended to $M > 2$.

**Theorem 1.** *Let $f_{NN} : \mathbb{R}^{D_1} \to \mathbb{R}^p$ be a NN function with random weights and $g : \mathbb{R}^{D_2} \to \mathbb{R}^p$ be a mapping function, and define an inner product between the representations $\hat{y} = f_{ICK}\left(\boldsymbol{x}^{(1)}, \boldsymbol{x}^{(2)}\right) = f_{NN}\left(\boldsymbol{x}^{(1)}\right) \cdot g\left(\boldsymbol{x}^{(2)}\right) = \boldsymbol{z}^{(1)^T} \boldsymbol{z}^{(2)}$. Then $f_{ICK}$ will converge in distribution to a GP in the infinite width limit a priori*

$$f_{ICK} \xrightarrow{d} \mathcal{GP}(0, K_1^{NN}(\boldsymbol{x}^{(1)}, \boldsymbol{x}^{(1)'}) K_2(\boldsymbol{x}^{(2)}, \boldsymbol{x}^{(2)'})), \tag{4}$$

*if $f_{NN}$ is a neural network with zero-mean i.i.d. parameters and continuous activation function $\phi$ which satisfies the linear envelope property in Equation 2 and $g$ includes the following deterministic kernel-to-latent-space transformation for all $1 \le i, j \le N$*

$$K_2(\boldsymbol{x}_i^{(2)}, \boldsymbol{x}_j^{(2)}) \approx \boldsymbol{z}_i^{(2)^T} \boldsymbol{z}_j^{(2)} = g(\boldsymbol{x}_i^{(2)})^T g(\boldsymbol{x}_j^{(2)}), \tag{5}$$

*where $K_1^{NN}$ is a NN-implied kernel and $K_2$ is any valid kernel of our choice.*

To prove Theorem 1, we first state following lemma.

**Lemma 2.** *For latent representations $\boldsymbol{z}_i^{(1)}$ and $\boldsymbol{z}_j^{(1)}$ extracted from different data points $\boldsymbol{x}_i$ and $\boldsymbol{x}_j$ where $i \neq j$, the interactions between different entries of $\boldsymbol{z}_i^{(1)}$ and $\boldsymbol{z}_j^{(1)}$ can be reasonably ignored. In other words, let $\theta^{(1)}$ be the parameters of the neural network which takes in $\boldsymbol{x}^{(1)}$ and outputs $\boldsymbol{z}^{(1)}$, we have $\mathbb{E}_{p(\theta^{(1)})}[z_{ik}^{(1)} z_{jl}^{(1)}] = 0$ for all $k \neq l$.*

A detailed proof of Lemma 2 is provided in Appendix A. With Lemma 2, let $\boldsymbol{\Theta} = \left\{ \boldsymbol{\theta}^{(1)}, \boldsymbol{\theta}^{(2)} \right\}$ represent the parameters of ICK, we can calculate the covariance between $\hat{y}_i$ and $\hat{y}_j$ for different data indices $i \neq j$:

$$\text{cov}(\hat{y}_i, \hat{y}_j)$$
$$= \mathbb{E}_{p(\boldsymbol{\Theta})}[\hat{y}_i \hat{y}_j] - \mathbb{E}_{p(\boldsymbol{\Theta})}[\hat{y}_i] \mathbb{E}_{p(\boldsymbol{\Theta})}[\hat{y}_j] \tag{6}$$
$$= \mathbb{E}_{p(\boldsymbol{\Theta})}\left[ \left( \sum_{k=1}^p z_{ik}^{(1)} z_{ik}^{(2)} \right) \left( \sum_{k=1}^p z_{jk}^{(1)} z_{jk}^{(2)} \right) \right] \tag{7}$$
$$= \mathbb{E}_{p(\boldsymbol{\Theta})}\left[ \sum_{k=1}^p \sum_{l=1}^p z_{ik}^{(1)} z_{jl}^{(1)} z_{ik}^{(2)} z_{jl}^{(2)} \right] \tag{8}$$
$$= \mathbb{E}_{p(\boldsymbol{\Theta})}\left[ \sum_{k=1}^p z_{ik}^{(1)} z_{jk}^{(1)} z_{ik}^{(2)} z_{jk}^{(2)} \right] \tag{9}$$
$$= \sum_{k=1}^p \mathbb{E}_{p(\boldsymbol{\theta}^{(1)})}\left[ z_{ik}^{(1)} z_{jk}^{(1)} \right] \mathbb{E}_{p(\boldsymbol{\theta}^{(2)})}\left[ z_{ik}^{(2)} z_{jk}^{(2)} \right] \tag{10}$$
$$\approx K_1^{\text{NN}}\left( \boldsymbol{x}_i^{(1)}, \boldsymbol{x}_j^{(1)} \right) \sum_{k=1}^p \mathbb{E}_{p(\boldsymbol{\theta}^{(2)})}\left[ z_{ik}^{(2)} z_{jk}^{(2)} \right]. \tag{11}$$

Here, from Equation 6 to Equation 7, we use the statement $\mathbb{E}_{p(\boldsymbol{\theta}^{(1)})}[z_{ik}^{(1)}] = 0$ in Section 3.3 and $\boldsymbol{\theta}^{(1)} \perp\!\!\!\perp \boldsymbol{\theta}^{(2)}$, which leads to $\mathbb{E}_{p(\boldsymbol{\Theta})}[\hat{y}_i] = \mathbb{E}_{p(\boldsymbol{\Theta})}[\hat{y}_j] = 0$. From Equation 8 to Equation 9, we get rid of all the cross terms using Lemma 2 and $\boldsymbol{\theta}^{(1)} \perp\!\!\!\perp \boldsymbol{\theta}^{(2)}$. Specifically, we have $\mathbb{E}_{p(\boldsymbol{\Theta})}[z_{ik}^{(1)} z_{jl}^{(1)} z_{ik}^{(2)} z_{jl}^{(2)}] = \mathbb{E}_{p(\boldsymbol{\theta}^{(1)})}[z_{ik}^{(1)} z_{jl}^{(1)}] \mathbb{E}_{p(\boldsymbol{\theta}^{(2)})}[z_{ik}^{(2)} z_{jl}^{(2)}] = 0$ for all $k \neq l$. From Equation 9 to Equation 10, we again make use of $\boldsymbol{\theta}^{(1)} \perp\!\!\!\perp \boldsymbol{\theta}^{(2)}$. From Equation 10 to Equation 11, we use the statement $\mathbb{E}_{p(\boldsymbol{\theta}^{(1)})}[z_{ik}^{(1)} z_{jk}^{(1)}] \approx K^{\text{NN}}(\boldsymbol{x}_i^{(1)}, \boldsymbol{x}_j^{(1)})$ from Section 3.3. If the kernel-to-latent-space transformation in $g(\cdot)$ is *deterministic*, we can remove the expectation sign from the summation term in Equation 11 and the covariance can be further expressed as

$$\begin{aligned} \text{cov}(\hat{y}_i, \hat{y}_j) &\approx K_1^{\text{NN}}(\boldsymbol{x}_i^{(1)}, \boldsymbol{x}_j^{(1)})(\boldsymbol{z}_i^{(2)^T} \boldsymbol{z}_j^{(2)}) \\ &= K_1^{\text{NN}}(\boldsymbol{x}_i^{(1)}, \boldsymbol{x}_j^{(1)}) K_2(\boldsymbol{x}_i^{(2)}, \boldsymbol{x}_j^{(2)}), \end{aligned} \tag{12}$$

which means that $\hat{y}$ approximately follows a GP with a multiplicative composite kernel $K^{\text{comp}}(\boldsymbol{x}_i, \boldsymbol{x}_j) = K_1^{\text{NN}}\left( \boldsymbol{x}_i^{(1)}, \boldsymbol{x}_j^{(1)} \right) K_2\left( \boldsymbol{x}_i^{(2)}, \boldsymbol{x}_j^{(2)} \right)$ *a priori*. This serves as a consise proof of Theorem 1. We also give a more detailed proof in Appendix B.

## 4.2 Kernel-to-latent-space Transformation

We now show how we can construct an appropriate mapping $g(\cdot)$ that approximately satisfies the assumed form of equation 5 and is used in the derivation of ICK from equation 11 to equation 12. Here we adopt two methods, Nyström approximation and Random Fourier Features (RFF), to map the kernel matrix into the latent space. Below, we elaborate the formulations for both methods, and give the methods and results for the Nyström method. The results of RFF will be presented in Appendix C. We name our framework with N*y*ström method and *r*andom Fourier Features ICK*y* and ICK*r*, respectively.

According to Yang et al. (2012), the Nyström method will yield much better performance than RFF if there exists a large gap in the eigen-spectrum of the kernel matrix. This phenomenon is mainly caused by how these two methods construct their basis functions. In particular, the basis functions used by RFF are sampled from a Gaussian distribution that is independent from the training examples, while the basis functions used by the Nyström method are sampled from the training samples so they are data-dependent. In our synthetic data experiments, we train our ICK framework using a batch size of 50. The eigenvalues of the kernel matrices computed from the first 4 batches of the synthetic data set are displayed in Figure 3. It can be observed that

the first few eigenvalues of the kernel matrix are much larger than the remaining eigenvalues. Namely, we observe a large gap in the eigen-spectrum of the kernel matrix, and Nyström method does generalize much better than RFF in our experiments.

### 4.2.1 Nyström Approximation

The main idea of Nyström approximation (Williams & Seeger, 2000) is to approximate the kernel matrix $\boldsymbol{K} \in \mathbb{R}^{N \times N}$ with a much smaller low-rank matrix $\boldsymbol{K}_q \in \mathbb{R}^{q \times q}$ where $q \ll N$ so both the computational and space complexity of kernel learning can be significantly reduced, yielding

$$\boldsymbol{K} \approx \hat{\boldsymbol{K}} = \boldsymbol{K}_{nq} \boldsymbol{K}_q^{-1} \boldsymbol{K}_{nq}^T. \tag{13}$$

The entries of $\boldsymbol{K}_q$ and $\boldsymbol{K}_{nq}$ can be calculated as $(\boldsymbol{K}_q)_{ij} = K(\hat{x}_i, \hat{x}_j), i, j \in \{1, 2, ..., q\}$ and $(\boldsymbol{K}_{nq})_{ij} = K(x_i, \hat{x}_j), i \in \{1, 2, ..., N\}, j \in \{1, 2, ..., q\}$, respectively. $x$ represents the original data points and $\hat{x}$ represents pre-defined inducing points (or pseudo-inputs (Snelson & Ghahramani, 2005)). In our study, these inducing points are chosen by defining an evenly spaced vector over the range of original data points. By performing Cholesky decomposition $\boldsymbol{K}_q^{-1} = \boldsymbol{U}^T \boldsymbol{U}$, where $\boldsymbol{U} \in \mathbb{R}^{q \times q}$, $\hat{\boldsymbol{K}}$ is expressed as

$$\begin{aligned} \hat{\boldsymbol{K}} &= \boldsymbol{K}_{nq} \boldsymbol{K}_q^{-1} \boldsymbol{K}_{nq}^T \\ &= \boldsymbol{K}_{nq} \boldsymbol{U}^T \boldsymbol{U} \boldsymbol{K}_{nq}^T = (\boldsymbol{U} \boldsymbol{K}_{nq}^T)^T (\boldsymbol{U} \boldsymbol{K}_{nq}^T). \end{aligned} \tag{14}$$

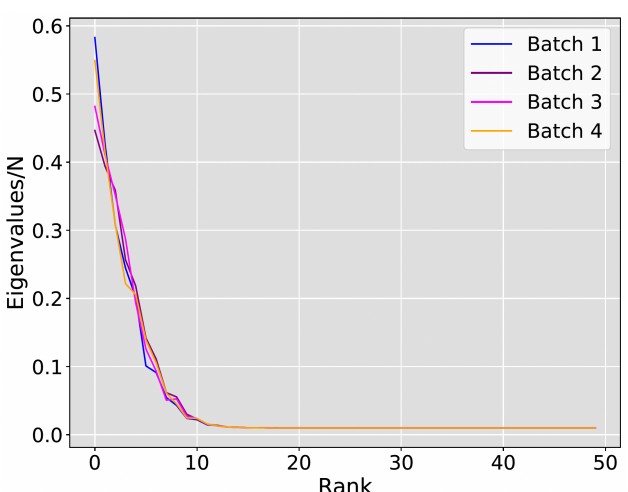

Figure 3: Eigenvalues of the kernel matrix computed from the first 4 batches of training data where $N$ is the total number of data points.

Therefore, if we set $q = p$, then we can use $\boldsymbol{z}_i \triangleq \boldsymbol{U}(\boldsymbol{K}_{np}^T)_{:,i}$ as a kernel-to-latent-space transformation because each element in $\boldsymbol{K}$ approximately satisfies equation 5 as stated in Theorem 1: $K(x_i, x_j) = K_{ij} \approx \hat{K}_{ij} = \boldsymbol{z}_i^T \boldsymbol{z}_j$. Conveniently, modern deep learning frameworks can propagate gradients through the Cholesky operation, making it straightforward to update the kernel parameters with gradient methods. Note that as we increase the number of inducing points $p$, the approximation error between $\boldsymbol{K}$ and $\hat{\boldsymbol{K}}$ decreases. However, it is not recommended to set $p$ very large as updating the Cholesky decomposition requires $\mathcal{O}(p^3)$. The empirical impact of $p$ on computational time and performance is shown in Appendix D. In our experiments, only mild values of $p$ are necessary and the impact on computational is relatively small. The full training procedure of ICK$y$ with $M = 2$ is presented in Algorithm 1.

### 4.2.2 Random Fourier Features

Random Fourier Features (RFF) is another popular approximation method used for kernel learning (Rahimi & Recht, 2007). Unlike the Nyström method which approximates the entire kernel matrix, RFF directly approximates the kernel function $K$ using some randomized feature mapping $\phi : \mathbb{R}^{D_m} \to \mathbb{R}^{2d_m}$ such that $K\left(x_i^{(m)}, x_j^{(m)}\right) \approx \phi\left(x_i^{(m)}\right)^T \phi\left(x_j^{(m)}\right)$. To obtain the feature mapping $\phi$, based on Bochner's theorem, we first compute the Fourier transform $p(\omega)$ of kernel $K$

$$p(\omega) = \frac{1}{(2\pi)^{D_m}} \int_{-\infty}^{+\infty} e^{-j\omega^T \delta} K(\delta) d\delta, \tag{15}$$

where $\delta = x_i^{(m)} - x_j^{(m)}$. Then we draw $d_m$ i.i.d. samples $\omega_1, \omega_2, ..., \omega_{d_m}$ from $p(\omega)$ and construct the feature mapping $\phi$ as follows

$$\phi\left(x^{(m)}\right) \equiv d_m^{-1/2} \left[\cos\left(\omega_1^T x^{(m)}\right), ..., \cos\left(\omega_{d_m}^T x^{(m)}\right), \sin\left(\omega_1^T x^{(m)}\right), ..., \sin\left(\omega_{d_m}^T x^{(m)}\right)\right]. \tag{16}$$

---

**Algorithm 1** Implicit Composite Kernel-Nyström (ICK$y$)

---

**Input:** data $\boldsymbol{X} = \left\{ \boldsymbol{x}_i^{(1)}, \boldsymbol{x}_i^{(2)} \right\}_{i=1}^N$, targets $\boldsymbol{y} = [y_i]_{i=1}^N$, $f_{\text{NN}}, \boldsymbol{\theta}^{(1)}, K^{(2)}, \boldsymbol{\theta}^{(2)}$, learning rate $\beta$

Sample a total of $N_B$ minibatches $\{\boldsymbol{X}_B, \boldsymbol{y}_B\}_{B=1}^{N_B}$
**for** $B$ from 1 to $N_B$ **do**
    **for** $\boldsymbol{x}_i^{(1)}, \boldsymbol{x}_i^{(2)}$ in $\boldsymbol{X}_B, i = 1, ..., n_B$ **do**
        $\boldsymbol{z}_i^{(1)} = f_{\text{NN}} \left( \boldsymbol{x}_i^{(1)} \right)$
        Define inducing points $\hat{\boldsymbol{x}}_1^{(2)}, ..., \hat{\boldsymbol{x}}_p^{(2)}$
        Compute $\boldsymbol{K}_p$: $(\boldsymbol{K}_p)_{jk} = K^{(2)} \left( \hat{\boldsymbol{x}}_j^{(2)}, \hat{\boldsymbol{x}}_k^{(2)} \right)$
        Do Cholesky decomposition $(\boldsymbol{K}_p)^{-1} = \boldsymbol{U}^T \boldsymbol{U}$
        Compute $\boldsymbol{K}_{np}$: $(\boldsymbol{K}_{np})_{jk} = K^{(2)} \left( \boldsymbol{x}_j^{(2)}, \hat{\boldsymbol{x}}_k^{(2)} \right)$
        $\boldsymbol{z}_i^{(2)} = \boldsymbol{U} \left( \boldsymbol{K}_{np}^T \right)_{:,i}$
        $\hat{y}_i = {\boldsymbol{z}_i^{(1)}}^T \boldsymbol{z}_i^{(2)}$
    **end for**
    $\hat{\boldsymbol{y}}_B = \texttt{concat} \left( \hat{y}_1, ..., \hat{y}_{n_B} \right)$
    Compute loss $\mathcal{L} = \mathcal{L} \left( \boldsymbol{y}_B, \hat{\boldsymbol{y}}_B \right)$
    $\boldsymbol{\theta}^{(1)} \leftarrow \boldsymbol{\theta}^{(1)} - \beta \nabla_{\boldsymbol{\theta}^{(1)}} \mathcal{L}$
    $\boldsymbol{\theta}^{(2)} \leftarrow \boldsymbol{\theta}^{(2)} - \beta \nabla_{\boldsymbol{\theta}^{(2)}} \mathcal{L}$
**end for**
**Return:** Predictions $\hat{\boldsymbol{y}}$ and updated parameters $\boldsymbol{\theta}^{(1)}, \boldsymbol{\theta}^{(2)}$

---

Since $\phi \left( x^{(m)} \right) \in \mathbb{R}^{2d_m}$, we need to set $d_m = p/2$ when using RFF as a kernel-to-latent-space transformation. In addition, since RFF involves sampling from a distribution, the kernel parameters are thus not directly differentiable and we need to apply a reparameterization trick (Maddison et al., 2016) to learn those parameters. The full training procedure of ICK$r$ with $M = 2$ is presented in Algorithm 2.

### 4.3 Uncertainty Estimation

Since the motivation of ICK discussed in Section 4.1 is closely related to GPs, we present two distinct approaches for estimating uncertainty in the subsequent sections: ensembling method and direct calculation of posterior variance.

#### 4.3.1 Ensembling

One approach for estimating the uncertainty is to adopt a sample-then-optimize approach (Matthews et al., 2017) and construct a deep ensemble posterior approximation for ICK. There are several ways to enable GP posterior interpretation for a deep ensemble trained with SGD. Specifically, let the final layer of each baselearner NN be $C$-dimensional and denote the deep ensemble as $F = \{f_{n_e}\}_{n_e=1}^{N_e}$, where $f_{n_e}$ is a baselearner NN and $N_e$ is the total number of baselearners in the ensemble. After training, each $f_{n_e}$ can be viewed as i.i.d. samples from a multi-output GP posterior with kernel function $K_F$ in the infinite width limit and the ensemble $F$ thus represents $N_e$ independent draws from the GP posterior, where $K_F$

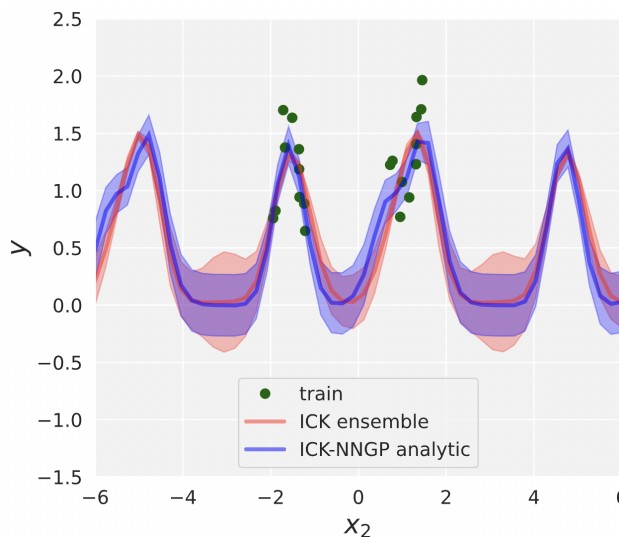

Figure 4: Predictive distribution from ICK$y$ ensemble and its GP posterior counterpart on a 1D regression task.

---

**Algorithm 2** Implicit Composite Kernel-RFF (ICK$r$)

---

**Input:** data $\boldsymbol{X} = \left\{\boldsymbol{x}_i^{(1)}, \boldsymbol{x}_i^{(2)}\right\}_{i=1}^N$, targets $\boldsymbol{y} = [y_i]_{i=1}^N$, $f_{\text{NN}}, \boldsymbol{\theta}^{(1)}, K_2, \boldsymbol{\theta}^{(2)}$, learning rate $\beta$

  Sample a total of $N_B$ minibatches $\{\boldsymbol{X}_B, \boldsymbol{y}_B\}_{B=1}^{N_B}$
  **for** $B$ from 1 to $N_B$ **do**
    **for** $\boldsymbol{x}_i^{(1)}, \boldsymbol{x}_i^{(2)}$ in $\boldsymbol{X}_B, i = 1, ..., n_B$ **do**
      $\boldsymbol{z}_i^{(1)} = f_{\text{NN}}\left(\boldsymbol{x}_i^{(1)}\right)$
      Compute the Fourier transform $p(\omega)$ of kernel $K_2$:
      $p(\omega) = \frac{1}{(2\pi)^{D_m}} \int_{-\infty}^{+\infty} e^{-j\omega^T \delta} K(\delta) d\delta$
      where $\delta = x_i^{(2)} - x_j^{(2)}$
      Draw $p/2$ i.i.d. samples $\omega_1, \omega_2, ..., \omega_{p/2}$ from $p(\omega)$.
      Construct feature mapping
      $\boldsymbol{z}_i^{(2)} = \sqrt{p/2}\Big[\cos\left(\omega_1^T x^{(2)}\right), ..., \cos\left(\omega_{p/2}^T x^{(2)}\right),$
      $\sin\left(\omega_1^T x^{(2)}\right), ..., \sin\left(\omega_{p/2}^T x^{(2)}\right)\Big]$
      $\hat{y}_i = \boldsymbol{z}_i^{(1)^T} \boldsymbol{z}_i^{(2)}$
    **end for**
    $\hat{\boldsymbol{y}}_B = \texttt{concat}\left(\hat{y}_1, ..., \hat{y}_{n_B}\right)$
    Compute loss $\mathcal{L} = \mathcal{L}\left(\boldsymbol{y}_B, \hat{\boldsymbol{y}}_B\right)$
    $\boldsymbol{\theta}^{(1)} \leftarrow \boldsymbol{\theta}^{(1)} - \beta\nabla_{\boldsymbol{\theta}^{(1)}}\mathcal{L}$
    $\boldsymbol{\theta}^{(2)} \leftarrow \boldsymbol{\theta}^{(2)} - \beta\nabla_{\boldsymbol{\theta}^{(2)}}\mathcal{L}$
  **end for**
**Return:** Predictions $\hat{\boldsymbol{y}}$ and updated parameters $\boldsymbol{\theta}^{(1)}, \boldsymbol{\theta}^{(2)}$

---

**Algorithm 3** ICK$y$ Ensemble

---

**Input:** data $\boldsymbol{X} = \left\{\boldsymbol{x}_i^{(1)}, \boldsymbol{x}_i^{(2)}\right\}_{i=1}^N$, targets $\boldsymbol{y} = [y_i]_{i=1}^N$, ensemble $F_{\text{ICK}y} = \{f_1, ..., f_{N_e}\}$ where each function in $F$ consists of $f_{\text{NN}}, K_2$ with parameters $\boldsymbol{\theta}^{(1)}, \boldsymbol{\theta}^{(2)}$, respectively

  **for** $s = 1, ..., N_e$ **do**
    Apply proper initialization strategy to $f_s$
    Perform Algorithm 1 on function $f_s$
  **end for**
**Return:** predictive mean $\hat{\boldsymbol{\mu}} = \frac{1}{N_e}\sum_{s=1}^{N_e} f_s(\boldsymbol{X})$, predictive variance $\hat{\boldsymbol{\sigma}}^2 = \frac{1}{N_e}\sum_{s=1}^{N_e}\left(f_s(\boldsymbol{X}) - \hat{\boldsymbol{\mu}}\right)^2$, and updated parameters $\boldsymbol{\theta}^{(1)}, \boldsymbol{\theta}^{(2)}$ for all functions $f$ in $F_{\text{ICK}y}$

---

can be either the *Neural Network Gaussian Process* (NNGP) kernel (Lee et al., 2017) if we make only the last layer trainable or the *Neural Tangent Kernel* (NTK) (He et al., 2020; Jacot et al., 2018) if we add a randomized and untrainable function to each baselearner.

This deep ensemble mechanism can be easily applied to ICK$y$. Specifically, the final prediction of ICK$y$ $\hat{y} = f_{\text{ICK}y}\left(\boldsymbol{x}^{(1)}, \boldsymbol{x}^{(2)}\right)$ can be viewed as a weighted sum of $\hat{y} = \sum_{k=1}^p \alpha_k z_k^{(1)}$ where $\alpha_k = z_k^{(2)} = g\left(\boldsymbol{x}^{(2)}\right)_k$. Hence, if we construct an ensemble of ICK$y$, $F_{\text{ICK}y} = \{f_1, ..., f_{N_e}\}$ (as shown in Algorithm 3), and $f_{\text{NN}}$ is appropriately initialized for all baselearners $f_s \in F_{\text{ICK}y}, s = 1, ..., N_e$, then all trained baselearners in $F_{\text{ICK}y}$ can be *approximately* viewed as i.i.d. posterior samples from a single-output GP with a multiplicative kernel $K_F K_2$. $K_F$ will either be an NNGP or an NTK and $K_2$ comes from our defined kernel. If the parameters of $f_{\text{NN}}$ (i.e. $\boldsymbol{\theta}^{(1)}$) are independently drawn from a zero-mean Gaussian and are fixed except the last layer (as specified by Lee et al. (2019)) for all baselearners in the ensemble, this corresponds to an NNGP. A detailed proof is in Appendix E.

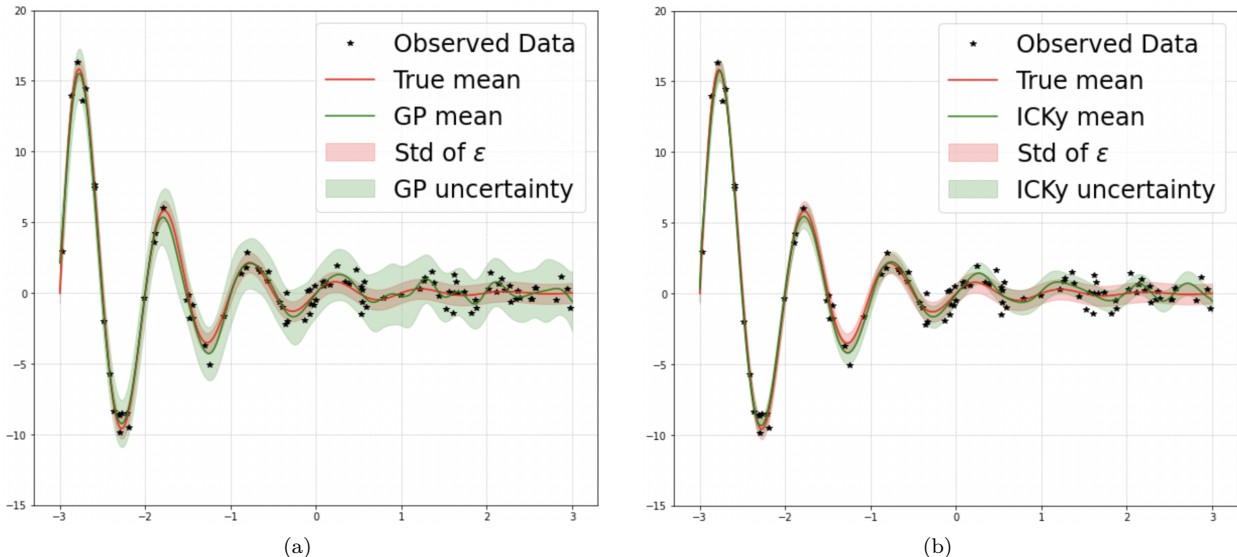

Figure 5: Plots of predicted mean and uncertainties for a sinusoidal function with exponential damping with noise $f(x) = e^{-x}\sin(2\pi x) + \epsilon$ using (a) an exact GP with RBF kernel and (b) ICK$y$ with RBF kernel. Here ICK$y$ predicts the uncertainty by directly calculating the covariance matrix $\boldsymbol{\Sigma}$ in Equation 17.

To verify our argument, we train an ICK$y$ ensemble $F_{\mathrm{ICK}y}$ containing 300 baselearners on the same 1D regression task as provided by He et al. (2020) using SGD optimizer with a learning rate of $5 * 10^{-4}$ and the mean-squared error (MSE) loss until convergence. We then compare the predictive distribution to its GP posterior counterpart based on the NNGP implementation in the *neural_tangents* (Novak et al., 2020) package. The weights for each baselearner are initialized to be drawn from an i.i.d. Gaussian with zero mean and variance equal to $\sigma_w/\sqrt{d_N}$ where $d_N$ is the width of the corresponding NN layer. The GP posterior has zero mean and kernel function $K_{\mathrm{NNGP}}K_2$, where $K_{\mathrm{NNGP}}$ is the corresponding NNGP kernel of $f_{\mathrm{NN}}$ and $K_2$ is the chosen kernel function for the mapping $g$ in the ICK$y$ ensemble in Figure 4. Here, $K_2$ is set to be an *exponential-sine-squared* kernel with period $T = 2\pi$. The predictive distribution of ICK$y$ ensemble is very close to the analytic GP posterior, demonstrating the *approximate* equivalence between the two models.

### 4.3.2 Computing Variance

Another approach to estimate the uncertainty is to directly compute the covariance matrix of the posterior distribution

$$\boldsymbol{\Sigma} = \boldsymbol{K}_{x^*x^*} - \boldsymbol{K}_{x^*x}\left(\boldsymbol{K}_{xx} + \sigma^2\boldsymbol{I}\right)\boldsymbol{K}_{xx^*}, \tag{17}$$

where $\boldsymbol{K}_{x^*x^*} = K(\boldsymbol{X}^*, \boldsymbol{X}^*)$ is the kernel matrix evaluated on the test dataset, $\boldsymbol{K}_{xx} = K(\boldsymbol{X}, \boldsymbol{X})$ is the kernel matrix evaluated on the training dataset, $\boldsymbol{K}_{x^*x} = K(\boldsymbol{X}^*, \boldsymbol{X})$ is the cross-covariance matrix evaluated on both training and test datasets, and $\boldsymbol{K}_{x^*x} = \boldsymbol{K}_{xx^*}^T$. Here $K$ refers to the composite kernel function implied by ICK. Note that if we get rid of $f_{\mathrm{NN}}(\cdot)$ and only keep $g(\cdot)$, then $K = K_2$ and the computation of $\boldsymbol{\Sigma}$ is straightforward. We believe the true uncertainty can be reasonably approximated by ICK$y$, as indicated by $\mathrm{diag}(\boldsymbol{\Sigma})$, while maintaining a reasonable computational complexity. To demonstrate this, we sample 100 data points from a sinusoidal function with exponential decay $f(x) = e^{-x}\sin(2\pi x) + \epsilon$ where $\epsilon \sim \mathcal{N}(0, 0.5)$ and see how an exact GP and ICK$y$ with radial basis function (RBF) kernel can capture the mean and uncertainty of this function by learning from these samples and directly calculating the covariance $\boldsymbol{\Sigma}$ according to Equation 17. In Figure 5, our results reveal that both GP and ICK$y$ accurately capture the mean function. However, ICK$y$ exhibits a prediction of uncertainty closer to the ground truth when compared to GP.

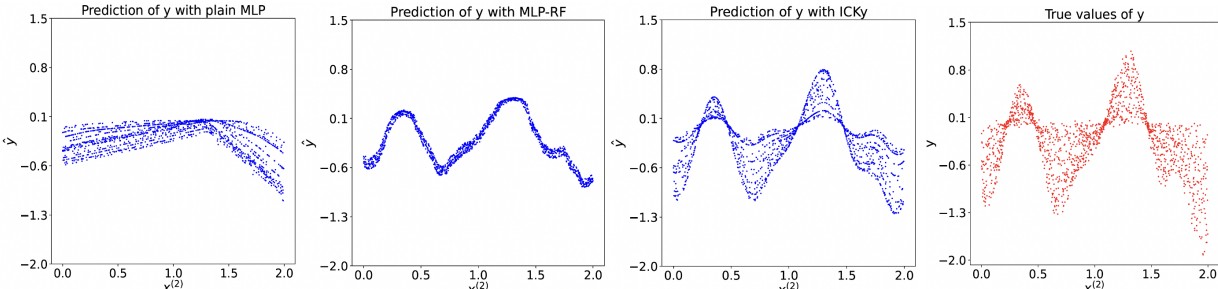

Figure 6: Prediction of $y \sim \mathcal{GP}(0, K_1K_2)$, where $\boldsymbol{x}^{(1)}$ is input to a linear kernel $K_1$ and $\boldsymbol{x}^{(2)}$ is input to a spectral mixture kernel $K_2$. We plot $\boldsymbol{x}^{(2)}$ against the predicted $y$. We show results from a plain MLP (left), MLP-RF (middle left), and ICK$y$ framework (middle right), and we compare to the true values of $y$ (right).

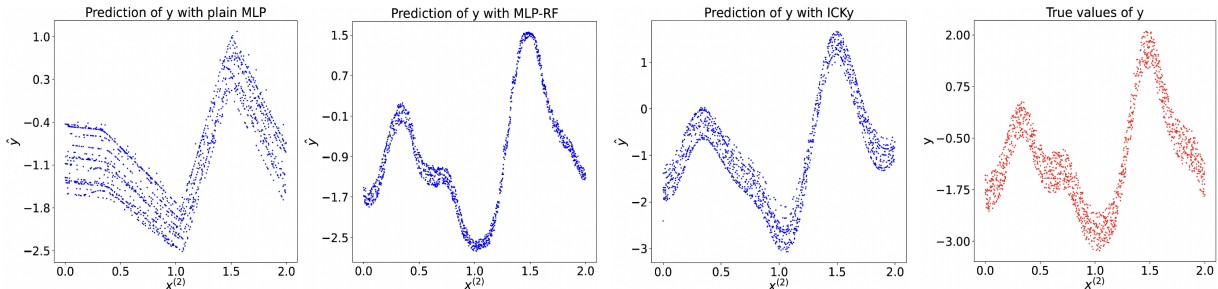

Figure 7: Prediction of $y \sim \mathcal{GP}(0, K_1 + K_2)$, where $\boldsymbol{x}^{(1)}$ is input to a linear kernel $K_1$ and $\boldsymbol{x}^{(2)}$ is input to a spectral mixture kernel $K_2$. We plot $\boldsymbol{x}^{(2)}$ against the predicted $y$. We show results from a plain MLP (left), MLP-RF (middle left), and ICK$y$ framework (middle right), and we compare to the true values of $y$ (right).

## 5 Experiments

We evaluate ICK$y$ on 6 different data sets: 2 synthetic datasets and 4 real-world datasets [1]. In all the experiments, our ICK$y$ framework only consists of 2 kernels (i.e. $M = 2$), one NN-implied kernel and one chosen kernel function with trainable parameters. The implementation details of all the experiments in this section are provided in Appendix F and the data accessibility and restrictions are provided in Appendix G.

### 5.1 Synthetic Data

#### 5.1.1 Case when $M = 2$

To verify that ICK$y$ can simulate sampling from a GP with *multiplicative kernel*, we create a synthetic data set $y \sim \mathcal{GP}(0, K_1K_2)$ containing 3000 data points where $\boldsymbol{x}^{(1)} \in [0, 1]$ is the input for the linear kernel $K_1$ and $\boldsymbol{x}^{(2)} \in [0, 2]$ is the input for a *spectral mixture* kernel (Wilson & Adams, 2013) $K_2$ with 2 components.

We compare ICK$y$ with two models: a multi-layer perceptron (MLP) applied to the concatenated features and a novel multi-layer perceptron-random forest (MLP-RF) joint model employed by Zheng et al. (2021), where MLP learns from $\boldsymbol{x}^{(1)}$ and RF learns from $\boldsymbol{x}^{(2)}$. We believe MLP-RF serves as a good benchmark model as it is a joint model with similar architecture to our ICK$y$ framework. To see how ICK$y$ simulates the spectral mixture kernel, we plot only $\boldsymbol{x}^{(2)}$ against the predicted value of $y$ as shown in Figure 6. As can be seen from the figure, plain MLP only captures the linear trend. MLP-RF only captures the mean of the spectral mixture components. In contrast, our ICK$y$ framework captures both the mean and the variance of the spectral mixture kernel. We also repeat this experiment by sampling from GP with an *additive kernel* and we obtain similar observations as shown in Figure 7.

---

[1]The code for all the experiments presented in this paper can be accessed at: `https://github.com/jzy95310/ICK`.

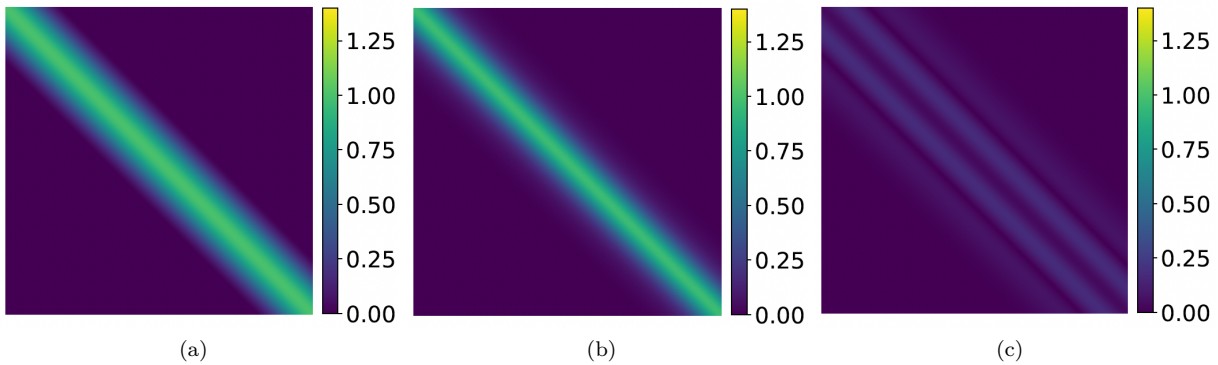

Figure 8: Visualization of (a) True matrix (b) estimated matrix by our ICK$y$ framework, and (c) absolute difference between the true and estimated matrix for the spectral mixture kernel

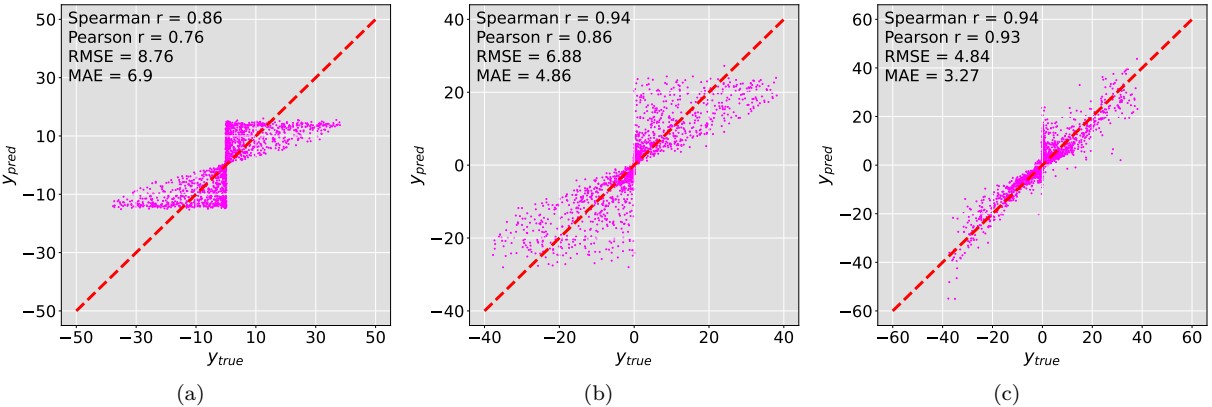

Figure 9: Scatter plots of the true values of $y$ against the predicted values of $y$ using our ICK$y$ framework with information from (a) one source $\hat{y} = f_{\mathrm{NN}}\left(x^{(1)}\right)$, (b) 2 sources $\hat{y} = f_{\mathrm{ICK}y}\left(x^{(1)}, x^{(2)}\right)$, and (c) 3 sources $\hat{y} = f_{\mathrm{ICK}y}\left(x^{(1)}, x^{(2)}, x^{(3)}\right)$.

We also examine whether ICK$y$ can retrieve the spectral mixture kernel in this task. After fitting the parameters of the spectral mixture kernel in ICK$y$, we compute the kernel matrix $\boldsymbol{K}_{\mathrm{ICK}y}$ using these learned parameters and compare it with the true kernel matrix $\boldsymbol{K}_{\mathrm{true}}$ by calculating the absolute difference between them as displayed in Figure 8. As can be observed, $\boldsymbol{K}_{\mathrm{ICK}y}$ and $\boldsymbol{K}_{\mathrm{true}}$ are similar and their absolute difference is relatively small, indicating that ICK$y$ can approximately retrieve the spectral mixture kernel.

### 5.1.2 Case when $M = 3$

We then construct another synthetic data set with 3000 data points where each input $\boldsymbol{x} = \left\{x^{(1)}, x^{(2)}, x^{(3)}\right\}$ contains information from 3 difference sources. The output $y$ is generated by $y = x^{(3)} \tanh\left(2x^{(1)} \cos^2\left(\pi x^{(2)}/50\right)\right) + \epsilon$ where $\epsilon$ is a Gaussian noise term. We process $x^{(1)}$ with a small single-hidden-layer NN, $x^{(2)}$ with an *exponential sine squared* kernel, and $x^{(3)}$ with a *radial-basis function* (RBF) kernel. Figure 9 shows the prediction results as we progressively add more sources of information into our ICK$y$ framework with corresponding kernel functions. It can be observed that ICK$y$ yields both smallest error and highest correlation with information from all 3 different sources. Hence, ICK$y$ works well with the $M = 3$ case and the regression performance is improved as we add in more information related to the target.

### 5.1.3  Cases when $M > 3$

To assess the robustness of ICK$y$ in handling diverse information sources, we conduct "stress tests" by evaluating its performance under varying numbers of information sources, i.e., when $M \in \{1, 2, 3, 4, 5\}$. In this analysis, we generate synthetic data following a methodology similar to that outlined in Section 5.1. Specifically, we model the data as $y \sim \mathcal{GP}\left(0, \prod_{i=1}^{M} K_i\right)$, where $K_1$, $K_2$, $K_3$, $K_4$, and $K_5$ are set to be the spectral

Table 1: Prediction errors of ICK$y$ on synthetic dataset generated by GP with multiplicative kernel, i.e., $y \sim \mathcal{GP}\left(0, \prod_{i=1}^{M} K_i\right)$.

|      | $M=1$  | $M=2$  | $M=3$  | $M=4$  | $M=5$  |
|------|--------|--------|--------|--------|--------|
| RMSE | 0.0334 | 0.0440 | 0.0466 | 0.2108 | 0.2560 |
| MAE  | 0.0266 | 0.0347 | 0.0353 | 0.1565 | 0.1790 |

mixture kernel, linear kernel, RBF kernel, exponential sine-squared kernel, and Matérn kernel with $\nu = 5/2$, respectively. As presented in Table 1, when $M > 3$, the performance of ICK$y$ starts to degrade. We posit that this degradation is primarily attributed to vanishing gradient, a phenomenon we will discuss further in the Discussion section.

### 5.2  Remote Sensing Data

We believe ICK$y$ will be particularly useful for remote sensing applications. In this experiment, we evaluate ICK$y$ on a remote sensing datasets where each data point $\boldsymbol{x} = \{x, t\}$ contains 2 sources of information: a three-band natural color (red-blue-green) satellite image $x$ as the high-dimensional information and the corresponding timestamp as the low-dimensional information. Our goal is to forecast the ground-level PM$_{2.5}$ concentration $\hat{y} = f(x, t)$ using both sources of information.

As PM$_{2.5}$ varies with time on a yearly basis, we use an *exponential-sine-squared* kernel with a period of $T = 365$ (days) to process the low-dimensional information $t$. The satellite images are processed with a CNN. The results of ICK$y$ are then compared with 2 benchmarks as shown in Table 2: a Convolutional Neural Network-Random Forest (CNN-RF) joint model (Zheng et al., 2020; 2021) (similar to the MLP-RF model in Section 5.1) and a *carefully designed* Seasonal CNN-RF model that maps $t$ into two new features, $\sin(2\pi t/365)$ and $\cos(2\pi t/365)$, to explicitly model seasonality.

We note that the inner product operation in ICK is similar in mathematical structure to attention-based mechanisms (Vaswani et al., 2017), which are popular in many deep learning frameworks. Therefore, we introduce 2 variants of ICK$y$ and compare them with another 4 benchmarks where the CNN is replaced by an attention-based mechanism based off a Vision Transformer (ViT) (Dosovitskiy et al., 2020) and DeepViT (Zhou et al., 2021) architectures. These models include ViT-RF, Seasonal ViT-RF, Seasonal DeepViT-RF, and Seasonal MAE-ViT-RF, where ViT is pretrained by a Masked Autoencoder (He et al., 2022). "Seasonal" here denotes that we use the transformed periodic representation defined previously. We note that we are unaware of Vision Transformers being used in this manner, and we

Table 2: Correlation and error statistics of ICK$y$ and other joint deep models with both convolutional and attention-based architectures on the PM$_{2.5}$ forecasting task. "S." denotes seasonal variants.

|                 | $R_{\text{Spear}}$ | RMSE   | MAE    | MSLL      |
|-----------------|--------------------|--------|--------|-----------|
| CNN-RF          | 0.00               | 194.63 | 185.83 | -         |
| ViT-RF          | 0.07               | 190.82 | 181.63 | -         |
| S. CNN-RF       | 0.62               | 53.36  | 39.38  | 96.77     |
| S. ViT-RF       | 0.66               | 56.45  | 41.73  | 14.69     |
| S. Deep-ViT-RF  | 0.65               | 56.36  | 42.46  | 17.63     |
| S. MAE-ViT-RF   | 0.67               | 53.87  | 40.78  | 31.09     |
| CNN-ICK$y$      | 0.62               | 53.46  | 39.76  | **10.92** |
| ViT-ICK$y$      | **0.68**           | 56.56  | 41.41  | 12208     |
| DeepViT-ICK$y$  | 0.66               | **52.41** | **35.93** | 38220  |

want to evaluate whether it is the model structure or the prior information that is improving the results. Hence, these attention-based models, while not a primary contribution, are novel and represent a good-faith effort to define models with similar forms. As displayed in Table 2, both standard CNN-RF and ViT-RF models yields very large errors on predicting PM$_{2.5}$ values. After including seasonality, CNN-RF performs significantly better and shares similar predictive performance with CNN-ICK$y$. After replacing CNN with ViT, we observe slight improvement in both RF-joint models (especially when pre-trained by MAE) and ICK$y$ variants. Among the models we present here, ViT-ICK$y$ and DeepViT-ICK$y$ achieve the highest correlation

Table 3: Prediction error of actual worker productivity on the test data set with ICK$y$ and other benchmark models.

|  | MSE $\downarrow (*10^{-3})$ | MAE $\downarrow (*10^{-2})$ |
|---|---|---|
| MLP | $20.16 \pm 1.26$ | $9.93 \pm 0.36$ |
| Cyclic MLP | $20.97 \pm 1.98$ | $10.16 \pm 0.77$ |
| Cyclic MLP-RF | $19.05 \pm 1.36$ | $9.70 \pm 0.48$ |
| DKL | $21.40 \pm 2.85$ | $11.14 \pm 0.94$ |
| ICK$y$, $T = 2$ | $3.43 \pm 1.42$ | $4.85 \pm 1.00$ |
| ICK$y$, $T = 7$ | $0.44 \pm 0.13$ | $1.43 \pm 0.15$ |
| ICK$y$, $T = 30$ | $\mathbf{0.31} \pm 0.09$ | $\mathbf{1.17} \pm 0.14$ |

Table 4: Root-mean-square-error (RMSE) and negative log-likelihood (NLL) of ICK$y$ with Matern-3/2 kernel and periodic kernel and two GP benchmarks on power consumption data.

|  | RMSE | NLL |
|---|---|---|
| Exact GP | $0.055 \pm 0.000$ | $-0.152 \pm 0.001$ |
| SVGP | $0.084 \pm 0.005$ | $-1.010 \pm 0.039$ |
| ICK$y$, Matérn-3/2 | $0.036 \pm 0.000$ | $-1.424 \pm 0.000$ |
| ICK$y$, periodic | $\mathbf{0.033} \pm 0.000$ | $\mathbf{-1.737} \pm 0.000$ |

and the smallest prediction error, respectively. To evaluate the uncertainty calibration capability of ICK$y$, we construct **ICK$y$ ensembles** by following Algorithm 2 and compare them with ensemble formulations of RF-joint model benchmarks. We use a criterion called Mean Standardized Log Loss (MSLL) as defined in Williams & Rasmussen (2006) to evaluate the uncertainty calibration. From Table 2, it can be observed that CNN-ICK$y$ ensemble achieves the smallest MSLL. In addition, we realize ViT-based ICK$y$ variants yield very large MSLL as they make predictions which are far from the true PM$_{2.5}$ labels with high confidence (i.e. small variance). We provide details and possible explanations in Appendix H.

## 5.3 Other Regression Datasets

To see if our ICK$y$ framework generalizes to other domains, we acquired two additional regression datasets from the UCI machine learning repository, one for predicting worker productivity (with 1197 samples and 15 features) and the other for predicting power consumption (with 2075259 samples and 9 features). For the worker productivity dataset, we employ ICK$y$ with an *exponential-sine-squared* kernel with different periods and compare them with benchmarks including MLPs (Al Imran et al., 2019) and Deep Kernel Learning (DKL) (Wilson et al., 2016b) as shown in Table 3. For the power consumption dataset, we use ICK$y$ with both an *exponential-sine-squared* kernel and a *Matérn 3/2* kernel and compare them with scalable exact GP (Wang et al., 2019) and stochastic variational GP (Hensman et al., 2013) as shown in Table 4. From the results, we observe that ICK$y$ outperforms all the benchmarks in both experiments (especially on the worker productivity data with a margin of almost one order of magnitude).

## 5.4 Adapting ICK for Classification

While regression tasks are the primary motivation for this paper, there are many ways to adapt GPR for classification tasks. For example, a binary classification model can be created by using a sigmoid (Williams & Barber, 1998) or probit link (Choudhuri et al., 2007) on the output of the GP. Succinctly, given a function $f(\boldsymbol{x}) \sim \mathcal{GP}(0, K(\boldsymbol{x}, \boldsymbol{x}'))$, the binary outcome probability is be given as $p(y = 1|f(x)) = \sigma(f(x))$. Likewise, a multiple classification model can be constructed by using a multi-output GP (or multiple GPs) and putting the outputs through a softmax function (Williams & Barber, 1998) or multinomial probit link (Girolami & Rogers, 2006). This strategy can be summarized by calculating $C$ different functions $f_c(\boldsymbol{x}) \sim \mathcal{GP}(0, K(\boldsymbol{x}, \boldsymbol{x}'))$ for $c = 1, ..., C$, where $C$ is the number of classes, and then calculating the class probabilities through a link function, $p(y|\boldsymbol{x}) = \text{softmax}([f_1(\boldsymbol{x}), f_2(\boldsymbol{x}), ..., f_C(\boldsymbol{x})])$.

This same logic can be used to construct a multiple classification model from ICK$y$. Succinctly, let $r_c = f_{\text{NN},c}(\boldsymbol{x}^{(1)}) \odot \boldsymbol{z}_c^{(2)}$, where $f_{\text{NN},c}$ denotes a neural network specific to the $c^{th}$ class and $\boldsymbol{z}_c^{(2)}$ represents the Nyström approximation specific to the kernel for the $c^{th}$ class. We note that often in a multi-output case the kernel parameters are shared, and so $\boldsymbol{z}_c^{(2)}$ would be an identical vector for each class. Then, the output probabilities for a data sample as $p(y|\boldsymbol{x}) = \text{softmax}([r_1, \ldots, r_C])$. This framework is learned with a cross-entropy loss.

To provide *proof-of-concept* of this multiple classification strategy, we implemented this model on a version of Rotating MNIST. In this task, a dataset was created by rotating each image in the dataset by a uniform

random value $\phi \in [0, 2\pi)$, thus creating a dataset with 60,000 images each with an associated rotation covariate $\phi$. We implemented the above multiple classification model with a periodic kernel over the rotation angle. This strategy yielded an accuracy of 92.3% on the validation data. This is lower than methods such as spatial transformers (Jaderberg et al., 2015) that report accuracy greater than 99%. However, those models explicitly use the fact that the information is simply rotated, whereas ICK is modeling a smooth transformation in the prediction function as a function of angle. This ICK classification model is much closer in concept to the way Rotating MNIST is used to evaluate unsupervised domain adaptation. While the evaluation strategy is different than our random validation set, the state-of-the-art accuracy on unsupervised domain adaption is 87.1% (Wang et al., 2020). Due to the lack of complete and fair comparisons, we are not claiming that ICK$y$ is state-of-the-art for classification, but ICK$y$'s classification model does seem reasonable and viable based upon this result.

## 6 Discussion

**Computational Complexity and Flexibility of ICK**  Compared to exact composite GP models which scale $\mathcal{O}(N^3)$, the training process of our ICK framework is more efficient as it leverages standard backpropagation to learn both the paramters of NN and the kernel function. Specifically, let $N_B$ be the number of data points in one mini-batch and $p$ be the number of inducing points for Nyström approximation. The computational complexity for $g(\cdot)$ is $\mathcal{O}(p^2 N_B + p^3)$. The computational complexity for the forward pass of $f_{\text{NN}}(\cdot)$ is $\mathcal{O}\left(N_B(k_1 k_2 + \cdots k_{L-1} k_L)\right)$ where $k_1, ..., k_L$ are the widths of each layer for a neural network of depth $L$. In addition, our ICK framework is more flexible compared to other joint models (i.e. BNNs and CNN-RF). Specifically, the BNNs of Pearce et al. (2020) cannot simulate complicated kernels such as the spectral mixture kernel we use in Section 5.1.

**Limitations**  As described in Section 5.1.3, ICK$y$ starts to yield higher prediction error when we have more than 3 different sources of information. This phenomenon is likely to be caused by vanishing gradient as we may need to multiply small numbers together due to the nature of inner product. Furthermore, in some cases, the properties of predictive posterior given by ICK ensemble are dominated by the neural network as shown in Appendix H. It can sometimes be challenging to choose an appropriate NN architecture to make sure it does not interfere with other specified kernels. In addition, as highlighted in Section 8.3.2 of Williams & Rasmussen (2006), it is important to note that the predictive variance using the Nyström method is not guaranteed to be positive. This can result in the kernel matrix no longer being positive-semidefinite, making the Cholesky decomposition infeasible. In such cases, we recommend either reducing the rank of the Nyström approximation or experimenting with alternative kernel functions.

**Broader Impacts**  We believe our framework is extensively applicable to regression problems in many fields of study involving high-dimensional data and multiple sources of information with perceptible trends, such as remote sensing, spatial statistics, or clinical diagnosis. Also, we are not aware of any negative societal impacts of our work.

## 7 Conclusion

This paper presents a novel yet surprisingly simple Implicit Composite Kernel (ICK) framework to learn from *hybrid* data containing both high-dimensional information and low-dimensional information with prior knowledge. We first analytically show the resemblance between ICK and composite GPR models and then conduct experiments using both synthetic and real-world data. It appears that ICK outperforms various benchmark models in our experiments with lowest prediction errors and highest correlations even with very limited data. Overall, we show that our ICK framework is exceptionally powerful when learning from *hybrid* data with prior knowledge incorporated, and we hope our work can inspire more future research on joint machine learning models, enhancing their performance, efficiency, flexibility, and generalization capability.

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
