# OpenReview forum: "Incorporating Prior Knowledge into Neural Networks through an Implicit Composite Kernel"
_TMLR — Accepted by TMLR_

### Review · Reviewer_tUmv · 2023-11-21

**Summary Of Contributions:**

The paper studies the problem of incorporating prior knowledge from multiple sources of information through implicitly defined Gaussian Process composite kernels. Instead of implementing the composite kernels explicitly, the paper proposes to use the inner product between two representations, one generated by a neural network and one by an efficient mapping function based on either Nyström approximation or random Fourier features. The manuscript then adopts a sample-then-optimize approach to approximate the full GP posterior distribution and investigates the efficacy of the approach on several synthetic and real-world datasets and learning contexts.

**Audience:**

Yes

**Claims And Evidence:**

Yes

**Requested Changes:**

n/a

**Strengths And Weaknesses:**

### Strengths

- In general, the paper is well-written. Background information and relevant literature are adequately covered. Discussions about the strengths and limitations of the approaches are clearly outlined.
- The technical developments of the work are rigorous, intuitive, and well-founded from existing works. Proofs are rigorous and correct, and algorithms and experiments are well-thought-out (several approaches to compute the kernel-to-latent-space transformation and uncertainty estimations; various neural network models and learning contexts).
- The paper introduces several major technical contributions to the problem of interest, including:
    - the introduction of the Implicit Composite Kernels (ICK) to approximate GP kernels
    - a theoretical proof (Theorem 1) that the ICK framework, under reasonable assumptions, is approximately equivalent to sampling from a Gaussian process regression (GPR) model with a composite kernel *a priori*
    - compared to exact composite GP models, the training process of the ICK framework is more efficient as it leverages standard back-propagation to learn both the parameters of NN and the kernel function

    The proposed approaches would be of broad interest in the machine-learning community from both theoretical and algorithmic viewpoints.

- The efficacy of the approaches is validated across several synthetic (2) and (4) real-world datasets and learning contexts (both regression and classification). The experiments show that the new approaches yield better performance on prediction and forecasting tasks, even with limited data.


### Weaknesses

- No weakness was noted.

---

### Review · Reviewer_dxRk · 2023-12-08

**Summary Of Contributions:**

In this paper, the authors develop a composite kernel framework that allows for combining the expressive nature of standard GP kernels (that can be used to model properties such as periodicity) with the flexibility of learning from large dimensions that is typically better suited to neural network architectures. Under this set-up, different subsets of the input domain can be treated independently by different kernels - these latent spaces are then combined via a chained inner product of the extracted representations. The full posterior distribution can then be approximated using either ensembling or analytically computing the covariance matrix of the posterior distribution. The suitability of the model is assessed on a variety of synthetic and real-world regression problems, where the proposed ICK framework outperforms competing models. An illustrative example of how the framework can be adapted to classification problems is also included in the paper, where an experiment is performed on the Rotating MNIST dataset.

**Audience:**

Yes

**Claims And Evidence:**

Yes

**Requested Changes:**

I highlighted the weaknesses of this work in the previous section. Although the paper is well-written overall, I find that the experiments and real-word examples are currently very limited in light of existing methods that already enable fairly similar composition of kernels. More ablation studies would also be beneficial for showcasing the extent to which the method is extendable to multiple dimensions.

**Strengths And Weaknesses:**

- The paper is well-written, and the core ICK architecture (along with its several variations) is clearly presented with the aid of neat figures and derivations. The problem statement is compelling, and I appreciated the sections describing how this paper relates to existing work on composite GPs and NNs incorporating prior knowledge.
- I appreciated that the authors took the time to show how ICK can be adapted to classification tasks, although it also come across as rushed - statements such as *"ICK’s classification model does seem reasonable and viable”* are somewhat inconclusive, and I interpreted this as an indication of inconclusive results.
- In the Limitations section, the authors indicate that ICK may not perform as well when there are a large number of information source. This claim alludes to an experiment that “stress tests” the model’s performance under this condition, but this isn’t explored in the current set of experiments.
- The paper’s motivation centres around datasets characterised by *"hybrid information”*, but the paper only contains one proper real-world example that demonstrates how this is suitable. The claim that *"ICK will be particularly useful for remote sensing applications”* is commendable, but limited in scope. I would appreciate more insight, and preferably additional experiments, that showcase other real-world domains where the use of ICK is expected to be beneficial.

---

> ### Author Response · Authors · 2024-01-15
> **Response to Reviewer dxRk**
>
> Thank you for your helpful comments. Regarding your questions and concerns, we have listed our answers below.
>
> **In the Limitations section, the authors indicate that ICK may not perform as well when there are a large number of information source. This claim alludes to an experiment that “stress tests” the model’s performance under this condition, but this isn’t explored in the current set of experiments.**
>
> To this end, we have included another subsection (i.e., Section 5.1.3) in the updated manuscript, where we explore the case when we have more than 3 sources of information. Specifically, we generate another synthetic dataset with a multiplicative GP $y \sim \mathcal{GP} \left( 0, \prod_{i=1}^{M} K_{i} \right)$ and report the prediction errors of ICKy as $M$ goes up. For further details, please refer to the updated manuscript and our implementation at: https://anonymous.4open.science/r/ICK-D4E6/experiments/synthetic_data/stress_test.ipynb.
>
> **The paper’s motivation centres around datasets characterised by "hybrid information”, but the paper only contains one proper real-world example that demonstrates how this is suitable. The claim that "ICK will be particularly useful for remote sensing applications” is commendable, but limited in scope.**
>
> While our primary focus is on datasets with “hybrid information”, we contend that ICKy can be effectively applied to single-modality datasets when there exists prior knowledge about the relationship between the target and specific input features. As illustrated in Sections 5.3 and 5.4, we demonstrate the feasibility of manually isolating these input features — such as the time index in Section 5.3 and the rotational angle of the image in Section 5.4 — and inputting them into $K_2$, while allocating the remaining features to $K_1$. We believe this strategy helps broaden the applicability of our proposed framework.
>
> Furthermore, we would like to point out that both datasets obtained from the UCI repository qualify as real-world datasets, though they require much less preprocessing compared to raw datasets in real-world settings.

---

### Review · Reviewer_b4Hq · 2024-01-05

**Summary Of Contributions:**

The paper proposes the implicit composite kernel method. This method is used to model datasets that contain both low- and high-dimensional data, where prior information is available. The authors attempt to draw a connection between their approximate method and Gaussian process regression. They evaluate the method on a synthetic task, showing that it can capture specific patterns. They also evaluate on a remote sensing dataset and two UCI regression tasks–demonstrating competitive performance compared to their benchmarks.

**Audience:**

Yes

**Broader Impact Concerns:**

None.

**Claims And Evidence:**

No

**Requested Changes:**

There are a number of missing citations that should be included in future versions of the paper.
- [1] should be cited on pages 2 and 3 for the connection between CNNs and GPs at infinite width.
- [2], [3] and [4] are other examples of combining NNs and GPs.
- It might be worth mentioning the line of work using iterative solvers to scale GPs to large datasets in the related work section. For example [5], even though this is cited in other places. See [6] also.

1. Novak, Roman, et al. "Bayesian deep convolutional networks with many channels are gaussian processes." arXiv preprint arXiv:1810.05148 (2018).
2. Hinton, Geoffrey E., and Russ R. Salakhutdinov. "Using deep belief nets to learn covariance kernels for Gaussian processes." Advances in neural information processing systems 20 (2007).
3. Wilson, Andrew G., et al. "Stochastic variational deep kernel learning." Advances in neural information processing systems 29 (2016).
4. Adlam, Ben, et al. "Exploring the uncertainty properties of neural networks' implicit priors in the infinite-width limit." arXiv preprint arXiv:2010.07355 (2020).
5. Wang, Ke, et al. "Exact Gaussian processes on a million data points." Advances in neural information processing systems 32 (2019).
6. Adlam, Ben, et al. "Kernel Regression with Infinite-Width Neural Networks on Millions of Examples." arXiv preprint arXiv:2303.05420 (2023).

**Strengths And Weaknesses:**

**Strengths**

The paper is written clearly and is easy to follow.


**Weaknesses**

[Connection between ICK and GPR]

The paper does not deliver fully on its stated contributions. The equivalence to Gaussian process regression is not handled correctly. The posterior of a finite-width NN trained with SGD is not a GP in general. Algorithm 3 makes no assumptions about the scale of the weights at initialization, the learning rate, the number of training steps, or the loss function. To be in the lazy training regime, assumptions are needed on all these.

[Experiments]

I'm quite confused by Section 5.3. Why were these datasets selected and not other UCI datasets, since they seem to contain no high-dimensional component? I also do not understand why the methods in Tables 2 and 3 were selected to compare against. Are they the current SotA? Are the numbers taken directly from the papers cited or were the methods reimplemented? I am also not able to find the numbers reported for DKL in the citation given. Even the comparison to Wilson in Table 3 seems to report a different number of datapoints (1,311,539).



**Minor issues**
- Novelty. ICK is a straightforward combination of existing methods, namely, featurizing inputs with a NN, multiplying kernels, and Nystrom/RFF approximation.
- Note that when using a Nyström approximation to the kernel, there is no guarantee that the variances will be positive. This is a known issue and should be discussed. See chapter 8 of [1].


1. Gaussian Processes for Machine Learning. Carl Edward Rasmussen and Christopher K. I. Williams. The MIT Press, 2006. ISBN 0-262-18253-X.

---

> ### Author Response · Authors · 2024-01-15
> **Response to Reviewer b4Hq**
>
> Thank you for your helpful comments. Regarding your questions and concerns, we have listed our answers below.
>
> **The posterior of a finite-width NN trained with SGD is not a GP in general. Algorithm 3 makes no assumptions about the scale of the weights at initialization, the learning rate, the number of training steps, or the loss function.**
>
> Indeed, a finite-width NN trained with SGD does not correspond to a GP posterior. However, in Section 4, we only claim Theorem 1 to hold **in the infinite width limit a priori**, which means “with the width of the NN approaching infinity” and “before training”. For Algorithm 3, we initialize each weight parameter to be drawn from an i.i.d. normal with zero mean and standard deviation $\sigma_w / \sqrt{d}$ where $d$ represents the width of the corresponding NN layer. We have also explicitly stated the learning rate, the number of training steps, and the loss function. Please see the highlighted texts in blue in the updated manuscript.
>
> **Why were these datasets selected and not other UCI datasets, since they seem to contain no high-dimensional component?**
>
> Based on our framework formulation, $K_2$ is used to capture the relationship between the low-dimensional information and the target variable. We chose the worker productivity and power consumption datasets due to the presence of more evident patterns between one specific input feature (namely, the time index) and the target, in comparison to other datasets available in the UCI repository. Despite the absence of a high-dimensional component in these datasets, we can manually separate the feature that exhibits a discernible relationship with the target and apply our methodology effectively. For further clarification, please refer to our response to the second question from Reviewer dxRk.
>
> **Benchmark methods and results in Tables 2 and 3 (which are now Tables 3 and 4 in the updated manuscript).**
>
> We consider the chosen benchmark methods as the state-of-the-art for deep networks and Gaussian Processes (GPs) applied to the worker productivity and power consumption datasets, respectively. The results of the benchmark methods, namely MLP, cyclic MLP, and DKL, presented in Table 3, have been re-implemented and are accessible at https://anonymous.4open.science/r/ICK_NNGP-17C5/experiments/Worker_productivity/GP_worker_productivity_data_experiment.ipynb. The benchmark results in Table 4 are taken directly from Wang et al. [1]. Additionally, we provide the total number of samples for the power consumption dataset, considering the size of the entire dataset immediately after downloading from the UCI repository, which we believe is accurate. Discrepancies with the numbers reported in [1] may arise because they probably report the number of training samples, not the total number of samples. The implementation details for this experiment are available at https://anonymous.4open.science/r/ICK-D4E6/experiments/uci_ml_repo/.
>
> **When using a Nyström approximation to the kernel, there is no guarantee that the variances will be positive. This is a known issue and should be discussed.**
>
> Thank you for pointing this out. We have included the discussion on the potential negative predictive variance of the Nyström method in the Limitation section. Please refer to the updated manuscript.
>
> **Missing citations that should be included in the future version of the paper.**
>
> Thank you for bringing these to our attention. We have included these missing citations in Section 2. Please see the highlighted texts in blue in the updated manuscript.

---

### Author Response · Authors · 2024-01-15
**Manuscript has been updated**

Dear reviewers,

Thank you for your time and effort in reviewing our paper. We have uploaded an updated version of our manuscript. For your convenience, we have highlighted all changes in blue and we will summarize the major changes below:

• We made some minor changes to Section 2 (Related Work) to include missing citations.

• We added training details of ICKy ensemble to Section 4.3.1.

• We included an extra experiment to "stress test" the number of modalities ICKy can handle, which can be found in Section 5.1.3.

• We included some further discussion of the limitations of Nystrom approximation in Section 6.

Please feel free to post more comments or questions in case our answers do not fully address your concerns.

Best regards,

Authors of Submission 1737

---

### Decision · Action_Editor_Y3Tx · 2024-02-23

**Recommendation:** Accept as is

**Comment:**

The paper introduces the Implicit Composite Kernel method, combining neural networks for high-dimensional data modeling with kernels for encoding prior knowledge. The subject of the paper is of importance to the ML community and is an area worthy of further investigation. However, as reviewers suggest, the paper would benefit from more comprehensive experiments and ablation studies.

**Audience:**

The proposed approaches would be of broad interest in the machine-learning community.

**Claims And Evidence:**

All reviewers agree the paper is well-written. One of them praised the paper for having "a strong technical component" and introducing "several major technical contributions." The others have concerns about the novelty and significance, stating "The novelty of the contributions is quite limited" and " [they] do not find the paper's results particularly novel or significant." They all agree the subject of the paper is of broad interest to the TMLR audience.

The paper receives one strong accept, one leaning accept and one leaning reject. The major concern for leaning accept/reject is (limited) novelty of the paper, which is not a primary acceptance criteria of TMLR.

I'm inclined to accept this paper. The topic is important and warrants further investigation, and the contributions outlined in the introduction appear well-supported.

---

> ### Author Response · Authors · 2024-02-28
>
> Dear Action Editor,
>
> We greatly appreciate your decision to accept our paper for publication in TMLR and your recognition of the significance of our contributions to the machine learning community!
>
> We have submitted the final camera-ready version of our paper, including a link to the code repository containing our model implementation and all the experiments discussed in the paper. Please feel free to leave a comment if you require any further information from us.
>
> Thank you once again for your support and consideration!
>
> Best regards,
>
> Authors of Submission 1737